# Shaping the Nascent Ribosome: AAA-ATPases in Eukaryotic Ribosome Biogenesis

**DOI:** 10.3390/biom9110715

**Published:** 2019-11-07

**Authors:** Michael Prattes, Yu-Hua Lo, Helmut Bergler, Robin E. Stanley

**Affiliations:** 1Institute of Molecular Biosciences, University of Graz, Humboldtstrasse 50/EG, A-8010 Graz, Austria; michael.prattes@uni-graz.at; 2Signal Transduction Laboratory, National Institute of Environmental Health Sciences, National Institutes of Health, Department of Health and Human Services, 111 T. W. Alexander Drive, Research Triangle Park, Durham, NC 27709, USA; yu-hua.lo@nih.gov

**Keywords:** AAA-ATPases, ribosome biogenesis, Rix7/NVL2, Rea1/Mdn1, Drg1/SPATA5, substrate processing, cryo-EM, small molecular weight inhibitors

## Abstract

AAA-ATPases are molecular engines evolutionarily optimized for the remodeling of proteins and macromolecular assemblies. Three AAA-ATPases are currently known to be involved in the remodeling of the eukaryotic ribosome, a megadalton range ribonucleoprotein complex responsible for the translation of mRNAs into proteins. The correct assembly of the ribosome is performed by a plethora of additional and transiently acting pre-ribosome maturation factors that act in a timely and spatially orchestrated manner. Minimal disorder of the assembly cascade prohibits the formation of functional ribosomes and results in defects in proliferation and growth. Rix7, Rea1, and Drg1, which are well conserved across eukaryotes, are involved in different maturation steps of pre-60S ribosomal particles. These AAA-ATPases provide energy for the efficient removal of specific assembly factors from pre-60S particles after they have fulfilled their function in the maturation cascade. Recent structural and functional insights have provided the first glimpse into the molecular mechanism of target recognition and remodeling by Rix7, Rea1, and Drg1. Here we summarize current knowledge on the AAA-ATPases involved in eukaryotic ribosome biogenesis. We highlight the latest insights into their mechanism of mechano-chemical complex remodeling driven by advanced cryo-EM structures and the use of highly specific AAA inhibitors.

## 1. Introduction to Eukaryotic Ribosome Biogenesis

Ribosomes are uniquely crafted to translate the genetic information encoded in mRNA into a polypeptide chain. To fulfil this fundamental cellular task, ribosomes have evolved to megadalton ribonucleoprotein complexes that are composed of one large (60S) and one small (40S) subunit. To cope with the demands for efficient translation, eukaryotic cells contain 200,000 to millions of ribosomes [1,2], each consisting of ~80 ribosomal proteins (r-proteins) and 4 ribosomal RNAs (rRNAs). Ribosome formation is therefore a high energy- and material-consuming process that is indispensable for cell growth and division. The tremendous complexity of the eukaryotic ribosomal assembly cascade is well illustrated by the number of involved components: around 250 additional factors are needed to facilitate ribosome production (recently reviewed in [3,4,5,6]). Although many maturation steps are highly conserved among eukaryotes, the formation of ribosomes in higher eukaryotes including humans requires many additional uncharacterized assembly factors [1,7]. Due to its fundamental importance, all stages of ribosome production are tightly regulated and linked to cell cycle progression and proliferation. Ribosome biogenesis is often upregulated in cancer cells, as it is a crucial determinant for fast growth, and therefore provides a promising target for anti-tumor chemotherapy [8,9,10,11,12,13,14,15]. By contrast, defects in ribosome biogenesis can lead to severe developmental diseases collectively termed “ribosomopathies” (reviewed in [1,3,16,17,18]).

Eukaryotic ribosome biogenesis has been best characterized in the yeast *Saccharomyces cerevisiae* and begins in the nucleolus (Figure 1) with the transcription of ribosomal DNA (rDNA) by RNA polymerases I and III (recently reviewed in [3,4,5,6,19]). RNA polymerase III transcribes the precursor of the 5S rRNA while RNA polymerase I produces a long 35S transcript. This 35S pre-rRNA includes all the other rRNAs (25S, 18S, and 5.8S) separated by internal transcribed spacers (ITS) and flanked by external transcribed spacers (ETS), as reviewed in [20]. Loading of the first assembly factors including small nucleolar ribonucleoprotein particles (snoRNPs) onto this 35S pre-rRNA scaffold occurs co-transcriptionally and forms the small subunit (SSU) processome, the earliest pre-ribosomal particle [21,22,23] (recently reviewed in [24]). After two processing steps in the 3′ ETS of the 35S pre-rRNA, an endonucleolytic cleavage at site A_2_ within ITS1 separates the maturation pathways of the two ribosomal subunits. From this step on, the precursors of the small and the large subunits go through separated maturation pathways. While the pre-40S particle is rapidly exported and finalized in the cytoplasm (recently reviewed in [19]), pre-60S particles undergo a cascade of maturation steps during their journey from the nucleolus through the nucleoplasm and into the cytoplasm (recently reviewed in [5,25]). Intriguingly, the three yeast AAA-ATPases directly shaping pre-ribosomes, Ribosome export 7 (Rix7), named nuclear VCP-like protein 2 (NVL2) in humans, ribosome export/assembly 1 (Rea1), designated Midasin 1 (Mdn1), and diazaborine resistance gene 1 (Drg1), termed spermatogenesis-associated 5 (SPATA5) are all associated with the formation of the large 60S subunit [26]. This is likely explained by the fact that the maturation cascade of the large subunit involves more individual steps and a larger number of additional assembly factors compared to small subunit maturation.

## 2. AAA-ATPases in Ribosome Biogenesis

### 2.1. How to Sculpt a Eukaryotic Ribosome Step by Step

From the early nucleolar stage onwards, distinct maturation intermediates (pre-ribosomal particles) are characterized and classified by their association with specific maturation factors. Pioneering work from several laboratories has unveiled structural snapshots of several of these pre-60S precursor particles from yeast (Figure 2) [27,28,29,30,31,32,33,34,35,36,37]. Defining the structure and composition of these particles has uncovered the metamorphosis of the large ribosomal subunit during maturation; however, many gaps still remain in the blueprints of ribosome assembly (recently reviewed in [3,4,5,6]).

During the cascade of maturation events, immature pre-60S particles undergo stepwise re-shaping of the pre-rRNA with its associated r-proteins from a flexible to a tightly packed and condensed globular shape [31,34]. With the help of the accumulating structural data, we are now beginning to understand how fundamental structural elements including the central protuberance (CP), the peptidyl transferase center (PTC) and the polypeptide exit tunnel (PET) are formed to obtain a functional large subunit. The rRNAs also serve as scaffolds to accommodate the sequentially incorporated r-proteins and transiently associating assembly factors. These assembly factors fulfil various functions at the pre-ribosome including post-transcriptional modifications, exo-nucleolytic trimming and endo-nucleolytic cleavage of the pre-rRNA, and large-scale structural rearrangements. Accordingly, ribosome formation is a progressive assembly process and the stepwise re-shaping is a strict prerequisite for the correct hierarchical incorporation of all r-proteins [5]. Ribosome formation can therefore be understood as a sophisticated sculpting process where the raw material is molded into shape by numerous consecutive working steps.

### 2.2. Order is Key: Molecular Machines Keep in Line

Many of the ribosomal maturation events are driven by energy-consuming enzymes including ATPases, kinases, GTPases, RNA helicases (DExD/H-box family) and snoRNPs [6,26,38,39,40]. The tight temporal and spatial coordination of all consecutive steps is crucial to ensure precise incorporation of all components and to establish the correct conformation of the rRNA. Some of the maturation factors bind only for a very short period, whereas others accompany the pre-ribosomes from the nucleolus all the way to the cytoplasm. At many stages, dissociation or active removal of assembly factors that are no longer needed is a strict requirement for the next step in the cascade (Figure 1). In fact, the binding surface for many maturation factors is often only prepared by the preceding maturation step and binding sites are initially occupied by other factors to prevent premature binding of late joining factors. These mechanisms serve as “quality control check points”, where maturation cannot proceed until the current step is correctly finished (reviewed in [25,41]). Indeed, with few exceptions, improperly assembled pre-ribosomal particles cannot proceed in the maturation cascade and are disassembled [25,42,43]. This strict hierarchical and cooperative assembly of each piece of the puzzle is a fundamental principle of eukaryotic ribosome formation [44,45]. AAA-ATPases, predestined for large-scale remodeling, have emerged as key players in this process by providing the mechanical force to actively strip maturation factors off pre-ribosomes. AAA-ATPases are also involved in the formation of snoRNPs (recently reviewed in [46]). However, this review will highlight the AAA-ATPases directly involved in the remodeling of pre-ribosomal particles.

Phylogenetically, AAA-ATPases belong to the P-loop NTPase family and can be classified in different clades and sub-families distributed over all kingdoms of life [47,48,49,50,51]. The well-characterized basic feature is the energy-driven enzymatic engine, the AAA domain (Figure 3A), which allows mechanical unfolding or degradation of proteins as well as remodeling of large multi-component complexes including eukaryotic pre-ribosomes [26,52,53,54,55]. Each AAA module is composed of two subdomains including the large αβ core domain (herein called AAA-L) comprised of alternating α-helices and β-strands followed by a small α-helical lid domain (AAA-S). The strict structural conservation of the catalytic core is contrasted by a huge variety of cellular functions achieved by different members of the AAA-family. A prime example for versatility is the non-catalytic N-terminal domain of the well-studied mammalian AAA-ATPase p97 which serves as a multifunctional interaction platform for numerous adaptor and cofactor proteins that target the AAA-ATPase to its substrates (e.g., ER- or mitochondria-associated proteins, as reviewed in [56]). With this modular system, the same core machinery formed by the AAA domains can be used to process a variety of different substrate proteins or complexes. Essential residues for binding and correct positioning of the nucleotide (Figure 3A,B) are provided by the Walker A motif (the P-loop), whereas the Walker B motif provides residues to coordinate a catalytic water molecule and Mg^2+^ ion to initiate hydrolysis of ATP [53,57].

A second characteristic of the AAA family is the conserved and structurally well-characterized oligomeric assembly. Most AAA-ATPases form doughnut-shaped hexamers (Figure 3C) with each monomer containing either one (type I) or two (type II) conserved AAA nucleotide binding domains (Figure 3D). Thus, one type II AAA hexamer can comprise up to twelve functional ATPase domains that act in a concerted manner to provide energy for mechanical remodeling. The catalytic center of the AAA nucleotide-binding pocket is formed at the interface between two adjacent monomers and is therefore only complete in the oligomeric state [58]. Nucleotide sensing residues, e.g., sensor I and the so-called arginine fingers (R-fingers), located in the second region of homology (SRH) of classical clade AAA-ATPases, allow transmission of the nucleotide binding state (ATP/ADP) from one protomer to the adjacent one which provides the basis for coordinated activity in all monomers of the ring [58]. The individual domains are often connected by highly conserved linkers that allow transmission of motions and inter-domain communication of the nucleotide binding state [59,60,61,62,63,64]. Aided by the flexibility of these linkers, the individual domains can run through large-scale conformational alterations during the ATPase cycle as well as positional changes relative to each other (movement of the N-terminal domain (NTD) relative to the D1 domain) or movements of the whole hexameric ring, e.g., movement of the D1 ring relative to the D2 ring (Figure 3D). The intricate choreography of these manifold conformational changes and movements of AAA hexamers is now being reinterpreted in the light of high-resolution cryo-EM structures depicting substrate-bound complexes which revealed asymmetric configurations during substrate processing (e.g., [65,66,67,68,69,70,71]).

### 2.3. General Modes of Substrate Processing

Although based on a conserved structural core, the substrate processing mechanisms of individual AAA-ATPases have evolved to perform different cellular tasks. Proteins including p97/Cdc48 [72,73], Hsp104 [65], Rix7 [68], archaeal VAT [66], and bacterial Clp [74] have been shown to act as translocases/unfoldases that can unfold a polypeptide chain by threading it through their central channel. Unfolding can serve as preparation for subsequent degradation of the polypeptide chain by proteases such as the proteasome [75,76]. By contrast, other AAA-ATPases act as segregases including mammalian N-ethyl maleimide sensitive factor (NSF) [77,78,79]. Segregation is achieved by applying mechanical tension to protein complexes (e.g., by a twist movement), resulting in the untangling of the individual components. In this case, no full unfolding of the substrate occurs which allows direct recycling for a following round of action. Therefore, whether the substrate handling mechanism of an AAA-ATPase involves partial or full unfolding of the polypeptide chain could be determined by the fate of the respective substrate (degradation versus recycling).

## 3. From the Nucleolus to the Cytoplasm: Mechanistic Insights and Cellular Functions of Rix7, Rea1, and Drg1

In the following sections we will discuss our current knowledge of the cellular function of the AAA-ATPases Rix7, Rea1, and Drg1 combined with the latest structural insights. This recent progress is largely based on high resolution cryo-EM structures, which have allowed a quantum leap in the understanding of the substrate processing mechanisms of these fascinating molecular machines.

### 3.1. Rix7: Remodeling of Nucleolar Pre-60S Particles

#### 3.1.1. Structural Insights into the Molecular Mechanism of Rix7

The first AAA-ATPase that steps in into the maturation cascade of the 60S subunit is the yeast protein Rix7 [80,81] with its human orthologue NVL2 [82]. Together with Drg1, Rix7 is closely related to the p97/Cdc48/NSF/Pex1 subfamily of AAA-ATPases [26]. As type II AAA-ATPase, each monomer of Rix7 consists of two AAA-domains (D1 and D2) accompanied by an additional unique N-terminal domain (Figure 4A). A comprehensive classification of all herein discussed AAA-ATPases including a complete sequence alignment to assess the degree of conservation can be found in [26].

Interestingly, partial deletion of the *S. cerevisiae* Rix7 NTD is not lethal, although it results in a severe growth defect [81,83]. Together with its unique fold, this raised the question of whether the Rix7 NTD also serves as a classical interaction platform and is involved in regulation of the ATPase activity as in other AAA-ATPases [84,85,86,87,88]. Yeast-two-hybrid experiments have shown that for binding of its proposed substrate protein Nsa1, Rix7 does not require the largest part of the NTD [81]. This is corroborated by co-immunoprecipitation assays that have shown that human NVL2 contacts its WD40 repeat containing substrate protein WDR74, the mammalian orthologue of Nsa1, via the D1 AAA domain [89]. Intriguingly, Rix7 contains an essential nuclear localization sequence (NLS) in the NTD between amino acids 174 and 202 [80]. NVL2, the human orthologue of Rix7, carries in total three proposed nuclear localization sequences in the NTD plus an additional nucleolar localization sequence (NoLS, RRKR motif) in the so-called unique domain (UD)-domain (NVL2 UD, amino acids 1–93), which is an extension of the NTD [90]. This UD-domain can interact with the nucleolar protein Nucleolin to mediate the interaction with (r)RNA and retention in the nucleolus [90,91]. Consistently, the yeast Nucleolin Nsr1 has been reported to bind nuclear localization sequences and is required for rRNA processing [92,93,94,95]. Thus, recruiting Rix7 to the nucleolus seems to be an important function of its NTD. In addition, a region of the NVL2 NTD interacts with the ribosomal protein L5 (uL18) [91], which might act as an adaptor on the pre-ribosome.

Except for an NMR structure of part of the NTD [90] and deposited coordinates of the isolated NVL D2 AAA domain (PDB ID 2X8A), structural information for Rix7/NVL2 has been lacking until recently. The cryo-EM reconstruction of the double Walker B mutant of *Chaetomium thermophilum* Rix7 was recently determined at 4.5 Å resolution [68], revealing the architecture of the asymmetrically stacked AAA rings depicted in Figure 4C. The cryo-EM scattering map of Rix7 lacks density for the entire NTD (amino acids 1–192), suggesting that this domain is flexible and dynamic (Figure 4B,C). A recent structure of Cdc48 in complex with its cofactor, the Ufd1-Npl4 heterodimer, suggests that both ATP and cofactor binding causes conformational changes within the Cdc48 NTD. Although the Cdc48 NTD is also stably folded in the absence of cofactors, it is plausible that Rix7 specific cofactors are required to stabilize the intrinsically disordered region of the NTD.

Aside from harboring a distinct NTD, another feature that distinguishes Rix7 from other type II ATPases is the presence of insertions following helix α7 in both the D1 and D2 domains [26]. Cryo-EM reconstruction has revealed that the α7 insertions from the D1 and D2 domains are structurally different from one another [68]. In the D1 insertion, helix α7 is extended and bends back towards the neighboring AAA protomer, suggesting a putative role for the post α7 extension in stabilizing the hexameric ring. By contrast, the α7 insertion in the D2 domain, which is only partially ordered, does not make extensive contacts with the neighboring protomer.

Similarly to other type II AAA-ATPases, the D1 and D2 domains of Rix7 stack on top of one another and form a central channel through the middle of the AAA domains. Surprisingly, in the Rix7 reconstruction the central channel harbors an unknown polypeptide that is gripped by five of the six Rix7 protomers. The presence of a polypeptide has been observed in several recent structures of ATP-hydrolysis-deficient mutants of AAA-ATPases or with the slowly hydrolyzable ATP-analogue ATPγS, suggesting that blocking ATP hydrolysis serves as a “substrate trap” [65,66,96]. There are two pore loops, pore loop 1 (PL-I) and pore loop 2 (PL-II), that line the central channel within each AAA domain of Rix7 (Figure 4B). A conserved aromatic-hydrophobic-glycine (most often Y/F-V-G) motif is typically found within the PL-I of AAA unfoldases such as in the regulatory particle of the proteasome or the ClpX component of the ClpXP proteases [97,98]. Even though Rix7 lacks the signature aromatic-hydrophobic-glycine motif within D1, it can engage a substrate throughout the entire central channel, suggesting that Rix7 functions as a molecular unfoldase. In the substrate-bound state, these pore loops are arranged in a spiral configuration around the central peptide. Interestingly, the D2 pore loops grip more of the peptide, which is likely to maximize the contact area within this domain (Figure 5A). ATP is present in the nucleotide binding pockets of all six protomers within the D1 AAA ring, whereas only four nucleotides have been observed in the nucleotide binding pockets of P1–P4 from the D2 AAA domain. Differences in the nucleotide binding states suggest that ATP-binding in D1 is necessary for substrate engagement while ATP hydrolysis in the D2 domain is important for substrate translocation. The sequence and structural differences of PL-I between D1 and D2 support the model that the D2 domain is the main ATP hydrolysis motor of Rix7 which drives translocation of the unfolding substrate [68].

Based on the asymmetrical arrangement and the different nucleotide binding states, the cryo-EM reconstruction supports the model that similarly to other type II AAA-ATPases, Rix7 remodels substrates through processive translocation (Figure 5B). Five of the six protomers (P1–P5) grab the substrate in the first step. The power stroke for substrate translocation is provided by ATP hydrolysis, which occurs sequentially in the D2 domain around the Rix7 hexamer. Each protomer takes a turn as the seam protomer, which is defined as the protomer detached from the substrate. By this hand-over-hand gripping mechanism, the polypeptide chain is step by step threaded through the channel and thereby unfolded. This model is further supported by point mutations of the D2 pore-loop residues which result in a lethal phenotype in yeast, confirming the critical role of the pore loops for Rix7 function in vivo [68]. Recent cryo-EM structures of substrate-engaged Cdc48 suggest a similar substrate processing mechanism with the D2 domain serving as the major player in ATP hydrolysis and substrate translocation [67,69]. Unlike the well-studied Cdc48/p97, which recognizes diverse ubiquitin modified substrates, there are still a large number of unanswered questions about substrate selection and processing by Rix7. For example, while Rix7 and Nsa1 are interaction partners it is unclear if Nsa1 is an authentic substrate of Rix7 or acts as an adaptor protein. Moreover, the mechanism of initial engagement and selection of substrate by Rix7 awaits further study.

#### 3.1.2. Remodeling of the Nucleolar Nsa1-Particle in Yeast

Consistent with its nucleolar localization, *S. cerevisiae* Rix7 is exclusively associated with late-nucleolar pre-60S particles that co-purify with the essential assembly factor Nsa1, named WDR74 in humans [81]. This Nsa1-associated pre-60S particle, which carries the 27SB pre-rRNA, already contains factors like Erb1, Nop7, Mak16, Rlp24, and Tif6 [31,81]. Corroborating a direct functional linkage between Nsa1 and Rix7, partial deletion or mutations in the Rix7 NTD synthetically enhance defects of mutations in Nsa1 and result in pre-rRNA processing and pre-ribosomal export defects [81]. Rix7 associates with the Nsa1-particle presumably rather transiently since it has been only found in sub-stoichiometric amounts [81]. However, since Rix7 is not detected on very early nucleolar Ssf1-particles or nucleoplasmic Rix1-particles, it proposedly fulfils its function before these particles transit from the nucleolus to the nucleoplasm. Interestingly, in stationary phase cells without active ribosome biogenesis, Rix7 is predominantly found in the nucleolus, whereas in exponentially growing cells, Rix7 distributes over the whole nucleus [80], potentially shuttling between the compartments.

Rix7 has been proposed to act at this early stage of ribosome biogenesis to catalyze the timely removal of Nsa1 from the pre-ribosome (Figure 6) [80,81]. Released Nsa1 can then be recycled to associate with nascent pre-ribosomes. Inactivation of Rix7 blocks Nsa1 release and leads to the mislocalization of Nsa1 to the cytoplasm [81]. However, it remains unclear if removal of Nsa1 is the exclusive function of Rix7. On the pre-ribosome, Nsa1 is part of a protein sub-complex, termed the Nsa1-module, formed by assembly factors Nsa1, Rrp1, Rpf1, and Mak16, which cluster at the solvent-exposed surface of the pre-ribosome [3,31]. Nsa1 associates with pre-60S particles at a rather early maturation stage, when many fundamental structural features of the 60S subunit including the 5S RNP, the central protuberance, the PTC, and the PET are still in an immature state [31]. In contrast, the so-called “foot”-structure, which will be removed in the nucleoplasm, is already well developed in Nsa1 particles [27,33,99]. In addition to changes in the protein composition and conformation, the domain organization of the 25S rRNA is not fully established. As revealed by successive snapshots of the subsequent maturation stages, the initially rather flexible individual domains of the 25S rRNA will go through massive conformational rearrangements to be correctly incorporated into a compact particle [31]. This incorporation resembles an origami-like folding procedure where intermediate states presumably need the stabilization by varying modules of assembly factors.

The whole Nsa1-module forms manifold interaction sites with protein and rRNA components of the pre-ribosome [31,34,37]. This module might function to stabilize the premature conformation of the pre-60S particle at this stage in preparation for further processing and rearrangement steps. Mak16 for example, which is also conserved in human cells, has been suggested to stabilize the 27SB pre-rRNA intermediate [100]. The Brix protein Rpf1 seems to function as a temporary placeholder for the correct formation of the PET and has to be removed prior to finalization of this structure [31]. Since all four components of the Nsa1-module are found on the same particle intermediates, the entire Nsa1-module could leave the pre-ribosome concertedly, although this has not yet been shown experimentally [3,4,31,34]. Nevertheless, the ATP-dependent dissociation of Nsa1 managed by Rix7 initiates conformational rearrangements of the late nucleolar particle to allow its efficient progression through the maturation cascade.

As a direct interaction partner, Nsa1 presumably recruits Rix7 to the pre-ribosome. At early stages, a conserved loop of the Nsa1/WDR74 β-propeller, which is needed for the interaction with Rix7/NVL2 [89], is not accessible, since it is shielded by interactions with the rRNA as well as Mak16 [31]. For Rix7 recruitment to the particle, a rearrangement of this area is necessary to make this interacting loop of Nsa1 available. This might determine the maturation stage at which Nsa1 has to be removed and therefore prevent premature release. Although the absence of Nsa1 is lethal for the cell, a failure to remove Nsa1 from the particle does not completely prevent the further maturation of pre-ribosomes. The majority of particles that still contain Nsa1 do not pass the quality control and are degraded. A sub-fraction is found in polysomes showing that Nsa1 is still present on mature and translating ribosomes [81]; however, it is unknown if these Nsa1-containing mature ribosomes are fully functional. Kater and coworkers speculate that the progression of these aberrant pre-60S particles is possible due to the exposed position of Nsa1 on the solvent-accessible side of the 60S subunit that does not interfere with binding of the small subunit [31]. Nonetheless, the structural rearrangements that are associated with the release of the other components of the Nsa1-module (e.g., clearing the PET from Rpf1) strictly have to occur to produce translation-competent 60S subunits. It has not yet been shown experimentally if these aberrant particles contain other components of the Nsa1 module [81].

Intriguingly, Nsa1 is dispensable for growth in cells harboring mutations in Mak5, Nop1, Nop4, and Ebp2 (Δ*nsa1* suppressor mutants), while Rix7 is still essential [101]. This suggests that Rix7 has additional cellular functions besides releasing Nsa1 or that Nsa1 primarily serves as an adapter that is not required in these mutant strains. Remarkably, overexpression of both wildtype Rix7 and dominant-negative AAA mutant variants in the absence of Nsa1 affects 60S maturation [101]. Accordingly, Nsa1 cannot be the sole Rix7 binding site on the pre-ribosome; otherwise, overexpressed Rix7 would not have a dominant-negative effect in cells lacking Nsa1. Interestingly, in the presence of Nsa1, overexpression of wildtype Rix7 only has negative effects on cells carrying the suppressor mutations, but not on wildtype cells [101]. Pratte and Coworkers have hypothesized that Rix7 can bind to aberrant pre-60S particles that arise in these mutant strains and direct them into a clearance pathway, since overexpression of Rix7 enhances the observed 60S deficiency in these strains. Binding of Rix7 might therefore act as a quality control checkpoint to sort out incorrectly assembled particles.

#### 3.1.3. Conservation of Rix7 Function in Eukaryotes

In mammalian cells, knowledge about the role of NVL2 is fragmented and the exact cellular function is still unclear. The human NVL gene encodes two isoforms of different lengths which are predominantly localized in different cellular compartments [82,91]. The shorter NVL1 isoform is mainly found in the nucleoplasm, whereas full-length NVL2 with the additional UD-domain and the NoLS localizes to the nucleolus [90,91]. WDR74, the mammalian orthologue of Nsa1, also exists in two different isoforms, of which to date only the longer isoform 1 has been documented to interact with NVL2 [102]. There is strong evidence that NVL2 also plays a crucial and specific role in 60S maturation and pre-rRNA processing, but the exact step of intervention is under debate [91,102,103,104,105]. Consistent with the situation in yeast, mutations of the human Nsa1 orthologue WDR74 cause pre-rRNA processing defects which resemble those of the dominant negative NVL2 mutant and affect pre-60S maturation in the nucleolus [103]. As described, the β-propeller domain of WDR74/Nsa1 interacts with the D1 AAA domain of NVL2/Rix7 [89], pointing at a conserved function in mammalian cells. However, analogous to yeast Nsa1, the exact role of WDR74 in pre-rRNA processing and 60S maturation is still not clear. Single nucleotide polymorphisms (SNPs) in the NVL gene are associated with an increased risk for mental disorders (schizophrenia and depression) and it has also been suggested as a prognostic outlier gene to assess the metastatic risk of prostate cancer patients [106,107]. These medical implications make it even more important to clarify the involvement of NVL2 in ribosome biogenesis.

NVL2 has also been linked to the activity of the nuclear exosome via co-immunoprecipitation and yeast-two-hybrid interaction with the DExD/H-box RNA helicase DOB1/Mtr4 [102,104,105]. The exosome catalyzes processing and degradation events of various RNA species, including pre-rRNAs, snoRNAs, and mRNAs, and is recruited by specific adaptor proteins [108,109,110,111,112]. In yeast, the Mtr4-associated exosome is recruited by Nop53 in the nucleoplasm and plays a role in the processing of the 7S pre-rRNA, which corresponds to the 12S in human cells [111,113]. In addition, the Mtr4-containing exosome is also targeted to an earlier ribosome biogenesis stage through an interaction with the SSU protein Utp18 [111]. Nevertheless, there is no evidence that links yeast Rix7 to the Mtr4-associated exosome for processing of 7S to mature 5.8S rRNA. However, as described earlier, Rix7 could be involved in the clearance of faultily assembled pre-ribosomes [101], which also involves degradation of the rRNA precursors by the exosome [42]. Recent work has identified a new Mtr4 binding motif within the NTD of NVL2 from higher eukaryotic organisms. This motif is reminiscent of the motif found in Utp18 and Nop53 [114]. Analogous to Nop53 and Utp18, this short motif in NVL2 binds to the Arch domain of Mtr4 and competes for the same binding site [114]. Thus, vertebrate homologues of NVL2 are nuclear adapters for the Mtr4-associated exosome.

The connection to both the exosome and WDR74 has raised the question of at which stage(s) of ribosome biogenesis the action of NVL2 is required. Mutations in the NVL2 AAA domains cause decreased generation of the 32S pre-rRNA from the 45S/47S pre-rRNA, but also decreased processing of 12S pre-rRNA to 5.8S rRNA, suggesting defects at two different stages in the processing cascade which are potentially linked to the exosome [105]. It is not clear yet whether this mirrors a direct involvement of NVL2 at two different stages in the maturation pathway or if the observed early defect is the result of a feed-back mechanism based on failed recycling of WDR74 and therefore a secondary effect [105]. Hiraishi and coworkers suggest that release of WDR74 by NVL2 might act as a regulatory checkpoint to prevent premature processing of the pre-rRNA [103]. In this context WDR74 could also act as an adaptor protein that helps to form and recruit the DOB1/Mtr4-associated exosome complex to the pre-ribosome.

In summary, yeast Rix7 and human NVL2 are both involved in the maturation of the pre-60S subunit, but it is not fully resolved as to whether they fulfil exactly the same task and how they are linked to the activity of the exosome. Although the direct interaction with Nsa1/WDR74 seems to be a conserved feature, both Rix7 and NVL2 might exert additional functions of unclear conservation. Other eukaryotic Rix7/NVL2 orthologues in, for example, *Drosophila melanogaster*, *Caenorhabditis elegans*, and *Toxoplasma gondii* have been linked to pre-rRNA processing but also to other cellular processes including cell division, proliferation, and apoptosis [90,104,115,116]. It is unclear if the proposed functions of these different orthologues are conserved across eukaryotes.

### 3.2. Rea1: A Colossus among Giants

#### 3.2.1. Unique Structural Features of Rea1/Mdn1

At the transition between nucleolus and nucleoplasm, the AAA-ATPase Rea1 in *S. cerevisiae* intervenes in the maturation cascade of the 60S subunit [26,83,117,118]. Rea1 is conserved from yeast to plants and humans where it is designated Mdn1 due to its conserved substrate-interacting metal-ion-dependent adhesion site (MIDAS) domain [119]. Depletion, knock-down, or mutation of Rea1/Mdn1 in yeast [83,118] or higher eukaryotes [120,121,122] is either lethal or has strong effects on cell growth and early development stages, demonstrating its essential function.

In contrast to Rix7 and Drg1, Rea1/Mdn1 belongs to the dynein-type family of AAA-ATPases [119,123,124,125]. Although Rea1/Mdn1 contains six AAA modules (Figure 7), it is not an assembly of six individual monomers but consists of one huge single polypeptide chain (scRea1: 4910 amino acids, ~560 kDa) that includes all six AAA domains plus an additional specialized C-terminal tail [118]. A common architecture of six concatenated AAA domains is also found within the motor protein dynein; however, Rea1 and dynein show little (~15%) sequence identity.

The six AAA modules of Rea1/Mdn1 show differences in length and conservation (a detailed multiple sequence alignment of the individual Rea1/Mdn1 AAA domains is presented in [26]). These differences suggest that in contrast to other AAA hexamers composed of identical monomers, the six AAA modules of Rea1 have evolved individually to adopt different roles in the substrate-processing mechanism. Although no functional data are available for all six AAA domains of *S. cerevisiae* Rea1, the sequence of AAA6 does not contain a functional Walker B motif and therefore has to serve a structural role (in *Schizosaccharomyces pombe* both AAA1 and AAA6 lack a functional Walker B). In addition, in vitro release of the substrate Rsa4 from pre-ribosomal particles is prohibited by a Walker A mutation in the AAA2 domain (K659A) that prevents ATP binding, showing the same effect as a mutation in the substrate-binding domain [129]. A systematic mutational analysis of residues needed for ATPase activity (Walker A/B, R-finger) of all individual AAA modules of *S. pombe* Mdn1 (spMdn1) has confirmed that only AAA2, AAA3, AAA4, and AAA5 have to be able to bind and/or hydrolyze ATP to support cell growth [130]. Moreover, AAA1 lacks the conserved R-finger residues which would normally be responsible for the sensing of the nucleotide bound in the neighboring AAA domain in the ring (AAA6) and coordinate hydrolysis. It remains an open question as to which way the individual domains exactly contribute to ATP-dependent substrate release.

The first structural views of Rea1 came from negative stain electron micrographs of *S. cerevisiae* Rea1 bound to nascent 60S particles which identified a “tadpole-like structure” [117]. The head of the tadpole was found to contain the pre-60S particle while antibody labeling confirmed that the tail-like structure was Rea1. Subsequent 2D class averages derived from negative stain micrographs of Rea1 on its own have revealed that Rea1 has two structural components including a hexagonal ring structure made up of the AAA domains and an elongated tail [118]. 2D class averages of Rea1 have also hinted at the flexibility of Rea1 as the orientation of the tail with respect to the ring has been seen to vary amongst classes [118]. Moreover, additional negative stain micrographs of Rea1-bound pre-60S particles have suggested that Rea1 makes two contact points with the ribosome, one through its AAA ring and the other through the MIDAS domain at the tip of its tail [118]. Collectively, these early structural snapshots led to the initial model that Rea1 couples ATP hydrolysis with conformational changes within its elongated tail.

The first near-atomic resolution cryo-EM structures of Rea1/Mdn1 were published in 2018 and include a series of reconstructions of Rea1 homologues from both *S. cerevisiae* [127] and *S. pombe* [128] in different nucleotide bound states. These reconstructions allowed high resolution insight into the overall architecture of Rea1′s N-terminal AAA ring and flexible C-terminal tail. The short NTD of Rea1 lies at the interface between AAA1, AAA6, and the base of the linker domain. The NTD acts as a scaffold holding together the ring and linker at this “ring-linker junction” (Figure 7B). The six AAA domains of Rea1 are arranged in an asymmetric configuration around the ring. With the exception of AAA6, the first five AAA domains are structurally similar to one another. Comparable to other hexameric AAA assemblies, the AAA-S domains form an interface with the AAA-L subdomains of the neighboring AAA (for example, AAA3-S contacts AAA4-L (Figure 7C)). This pattern is only broken by AAA6, which lacks a canonical AAA-S subdomain.

A distinguishing feature of Rea1/Mdn1 is the presence of β-sheet insertions following helix 2 (H2) in each AAA-L subdomain. In AAA1-L, AAA3-L, and AAA5-L these insertions are small β-hairpins; however, in AAA2-L, AAA4-L, and AAA6-L these insertions are quite large and in addition to the β-hairpin they include α-helical extensions that protrude along the bottom of the AAA ring [127,128]. The H2 insertion in AAA6 forms a 4-helix bundle that is well-ordered and associates with the AAA1-L subdomain. Intriguingly, the α-helical bundle of the AAA2 H2 insertion (H2α) partially occupies the channel in the middle of the AAA ring, suggesting it may play an important auto-regulatory role by inhibiting ATP hydrolysis [127,128].

The extended linker of the protein which follows the AAA domains can be broken down into subdomains including the stem, middle, top, and D/E-rich components [127,128]. The α-helical stem serves as the connection point with the AAA ring at the ring linker junction (Figure 7B). The end of the stem forms the hinge point for the Rea1 tail. The following middle subdomain is primarily composed of α-helices, with one helix extending down along the stem subdomain to contact AAA6. The top subdomain is composed of three α-helical bundles arranged in a horseshoe-like pattern at the top of the middle domain [127]. The following D/E-rich region of Rea1 is not visible in any of the cryo-EM reconstructions, suggesting that it is highly dynamic and flexible. At the very C-terminus of Rea1 lies the MIDAS domain, which is important for engaging Rea1′s substrates Ytm1 and Rsa4. Density for the MIDAS domain is visible in the cryo-EM reconstructions of scRea1 captured in the presence of AMP-PNP with a truncation of the H2 insertion from AAA2 [127] and spMdn1 stalled with ATP and Rbin-1 [128]. In these reconstructions weak density for the MIDAS domain can be observed at the top of the AAA ring above AAA3 in both the spMdn1 and scRea1 structures (Figure 7D) and is consistent with the variable tail conformations observed in early 2D class averages [118]. The recent structures support the idea that nucleotide-induced conformational changes are important for MIDAS domain engagement with the AAA ring. At the pre-ribosome, engagement of the MIDAS domain with the ring brings it in close contact with the proposed substrate Rsa4 as a prerequisite for the remodeling reaction [127,128]. In addition, binding to the pre-ribosome could dislocate the AAA2 H2α insertion from the central channel to allow ATP hydrolysis [127,128].

A higher resolution view of the MIDAS domain has been recently obtained from a series of crystal structures of the Rea1 MIDAS domain from *C. thermophilum* [126]. These structures revealed that the Rea1 MIDAS domain contains three elements that are not found in traditional integrin MIDAS domains. The first element is a helix that provides structural support. The second element is a disordered loop that includes a unique NLS sequence while the third element undergoes a large conformational change upon ligand binding [126]. Complex crystal structures of the ctRea1 MIDAS domain with the ubiquitin-like or MIDAS-interacting domain (UBL/MIDO) domains of Ytm1 and Rsa4 revealed that upon UBL binding the third element transitions from an unstructured loop to a structured β-hairpin that provides an additional binding surface for the UBL domains of Ytm1 and Rsa4 [126]. Docking of the MIDAS-UBL structure into the coordinates of the spMdn1 cryo-EM reconstruction obtained in the presence of ATP and Rbin-1 [128] suggests that there is additional density for the MIDAS domain that is not accounted for in the crystal structures [126]. One possibility is that this density could be the disordered loop from the second element. This density is visible within the center of the AAA ring and hints towards the possibility that this loop is important to tether the MIDAS domain onto the AAA ring [126].

One thing that was surprising from the cryo-EM reconstructions of scRea1 was that different nucleotide binding states did not appear to cause large scale conformational changes within the linker domain. For example, the ADP and AMP-PNP states of scRea1 are highly similar with the AAA2 H2 insertion blocking the central channel in both reconstructions [127]. Deletion of the AAA2 insertion has not been found to cause large scale conformational changes but has been found to lead to the docking of the MIDAS domain onto the Rea1 AAA ring. Similarly, reconstructions of spMdn1 in the presence of AMP-PNP or ATP + Rbin-1 [128] have been found to have overall very similar organizations of the ring and linker regions. However, the spMdn1 structure determined in the presence of ATP + Rbin-1 has revealed conformational changes within the AAA ring that may be linked with Rea1 ATPase activity [128]. These conformational changes lead to the described displacement of the AAA2 H2 insertion from the central channel caused by a movement of the AAA subdomains and the binding of the MIDAS domain onto the AAA ring. The recent cryo-EM structures suggest that Rea1/Mdn1 does not function by long-range motions within the linker-like dynein but rather works by conformational changes within the AAA ring that lead to binding and displacement of the MIDAS domain [127,128]. Different linker conformations of scRea1 have been documented by negative staining [118]. Therefore, it remains unknown how the linker and its flexibility contribute to the release of Ytm1 and Rsa4. Moreover, Rea1 shares characteristics with unfoldases like Rix7 in that its central channel is occupied by a polypeptide derived from either the AAA2 H2 insert or the MIDAS domain [126]. While more work is needed to delineate the exact mechanism of Rea1-stimulated release of Ytm1 and Rsa4, recent structures have shed light on the conformational dynamics of Rea1.

#### 3.2.2. One Giant Ratchet for Two Jobs: Pre-60S Remodeling by Rea1 at Multiple Stages

Based on genetic and biochemical data, *S. cerevisiae* Rea1 has been proposed to act at two distinct maturation stages of pre-60S particles (Figure 8). First, it was suggested that Rea1 remodels Rix1-associated pre-60S particles in the nucleoplasm by triggering the release of the assembly factor Rsa4 [83,118]. Rea1 was later proposed to also act at an earlier stage in the nucleolus by catalyzing the release of the assembly factors Ytm1 and Erb1 from late-nucleolar Nug1-associated pre-60S particles as a prerequisite for transition into the nucleoplasm [131]. This involvement of Rea1 in the nucleolus should tightly follow the above-described remodeling step catalyzed by Rix7. Both maturation stages associated with Rea1 are phases of massive structural rearrangements of the pre-ribosomes [27,31] reviewed in [3,4,5,6].

#### 3.2.3. Step 1: Remodeling of the Transiting Pre-60S Particle

Early nucleolar particles from yeast contain the maturation factor Erb1, which forms a versatile binding platform for a multitude of proteins, including Ytm1 [31,132]. Together with Nop7, Ytm1 and Erb1 form a trimeric sub-complex which is required for correct trimming of the 27SA_3_ pre-rRNA intermediate, a precursor of the mature 25S rRNA [133,134,135,136]. Biochemical and yeast-two-hybrid data have shown that the UBL domain of Ytm1 interacts with the MIDAS domain of Rea1 [131,137]. In vitro incubation of Rea1 with Rix1-containing pre-60S particles purified from Rea1-depleted cells has been found to result in the ATP-dependent release of Ytm1 and Erb1 as well as to a lesser extent Nop7 [131]. This in vitro release is dependent on a physical interaction between Rea1 and Ytm1. Abolishing this interaction by mutating the UBL-domain of Ytm1 (Ytm1-E80A) has been seen to prohibit the release of these factors from the pre-ribosomes [131]. These observations led to the initial model that Rea1 is recruited via a direct interaction with Ytm1 to release the Ytm1-Erb1-Nop7 complex in an ATP-dependent manner from late nucleolar pre-60S particles. However, there is still no structure of a particle that shows binding of Rea1 at this stage. Therefore, the question remains as to what additional contacts on the pre-ribosomes (beside Ytm1) are needed for Rea1 to exert its remodeling activity.

Erb1 not only interacts with Ytm1 and Nop7 but has been described as a “multivalent binding hub” that coordinates the interaction with ribosomal rRNAs as well as multiple maturation factors [31,37,132,138,139,140]. Exposed on the surface, the C-terminal β-propeller of Erb1 thereby interacts with the β-propeller domain of Ytm1 [31,132,140]. A long N-terminal string of Erb1 is deeply threaded into the pre-ribosome to mediate a large-scale interaction network that presumably stabilizes the premature architecture of the particle at this stage. Consequently, the removal of Erb1 will result in major rearrangements of the pre-rRNA and concomitantly the whole pre-60S particle. Rea1 could provide the energy to drive this large-scale restructuring [31]. The rearrangements associated with the removal of Ytm1 and Erb1 are considered a prerequisite to correctly form the 60S structure surrounding the PET and recruit maturation factors to the PTC, as well as for the rotation of the so-called L1 stalk to a more mature conformation [5,31].

Nop7, the third component of this sub-complex, is associated with the characteristic “foot structure” which is formed by the ITS2 pre-rRNA together with associated maturation factors and which is needed for the correct processing of 27SB pre-rRNA [33,113]. The components of the foot structure (the “A3 factors” including the later joining Nop53) organize the recruitment of factors that initiate cleavage of the 27SB pre-rRNA and subsequent exo-nucleolytic trimming of the 7S and 25.5S pre-RNAs [113,132,141]. While Nop7, Erb1, and Ytm1 form a stable complex in vitro, they do not seem to leave the pre-60S particle at the same maturation step [131,134,136]. Structural investigations have revealed that Erb1 has to leave its place to allow binding of Nop53, which then recruits the nuclear Mtr4-associated exosome to process the 7S pre-rRNA [31,111]. Accordingly, Erb1 has to be released before the exosome can be recruited. Nop7, by contrast, is the last ITS2 binding factor still present when the exosome is engaged with trimming of the 7S pre-rRNA from the 3′ end [113,132]. Thus, Ytm1 and Erb1 leave pre-60S particles prior to Nop7. This is further supported by structures of nucleoplasmic pre-60S particles purified with Nog2 and Arx1 as bait proteins [33,142]. Finally, an earlier dissociation of Ytm1 and Erb1 was suggested by kinetic investigations using the Drg1 inhibitor diazaborine [143]. This nicely demonstrates that the described modules of maturation factors do not necessarily act as rigid entities but behave dynamically during the maturation process with distinct roles and association/dissociation kinetics.

In human cells, a conserved nucleolar protein assembly known as the PeBoW complex is formed by Pes1 (homologue of Nop7), Bop1 (Erb1), and WDR12 (Ytm1). The PeBoW complex likely plays an analogous role in ribosome biogenesis as in yeast, since it is essential for 5.8S and 28S rRNA formation and correct maturation of pre-60S particles [133,144,145,146,147,148,149,150]. Consistent with a role as a recruiting platform, the mammalian PeBoW complex also interacts with the DEAD-box helicase DDX27 needed for 47S rRNA 3′-end formation [1,147]. In *Arabidopsis thaliana*, it has also been reported that Mdn1/Rea1 MIDAS interacts with the UBL domain of the Ytm1 orthologue Pes2 [121]. These examples suggest that the involvement of Rea1/Mdn1 at this step in ribosome biogenesis is conserved among species from yeast to higher eukaryotes.

#### 3.2.4. Step 2: Remodeling of the Nucleoplasmic Rix1-Associated Pre-60S Particle

After progression of the pre-60S particle from the nucleolus to the nucleoplasm, yeast Rea1 steps into the ring a second time to release the maturation factor Rsa4 from a late nucleoplasmic particle [83,118]. A prerequisite for stable association of Rea1 with the pre-ribosome at this stage is the presence of the salt-stable Rix1-Ipi3-Ipi1 sub-complex [27,117,118]. Rix1 appears to function as one of the main mediators between Rea1 and the pre-60S subunit because abolishing the direct interaction between Rea1 and Rix1 prevents recruitment of Rea1 to the particle [118]. In addition, Rix1 proposedly helps to dislocate the auto-inhibiting AAA2 H2α insertion to render Rea1 catalytically active [127,128]. An analogous complex in mammalian cells is formed by PELP1 (Rix1), TEX10 (Ipi1), and WDR18 (Ipi3). This complex has been reported to be needed for functional 28S rRNA maturation and transport of pre-60S particles from the nucleolus to the nucleoplasm [151,152]. An analogous physical and functional connection has also been described in *A. thaliana* between atMdn1 and the Rsa4-orthologue Nle1 (Notchless 1) [153,154,155]. These findings underline the conservation of these steps in the ribosomal maturation cascade in higher eukaryotic cells.

The nucleoplasmic pre-60S maturation stage connected to Rea1 is associated with a hallmark remodeling event of the pre-ribosome, the rotation of the 5S RNP [32]. The 5S RNP is a sub-complex formed by 5S rRNA and the ribosomal proteins L5 (uL18 according to the new nomenclature [156]) and L11 (uL5) [32,157,158]. In mammalian cells, an excess of free 5S RNP not associated with pre-ribosomal particles acts as a cellular sensor for disturbed ribosome biogenesis and nucleolar stress and results in the activation of p53 via Mdm2 [159,160]. This is one of many nodes connecting ribosome biogenesis to the regulation of cell growth and proliferation (recently reviewed in [1,3]).

As demonstrated in yeast, the conserved Rpf2-Rrs1 heterodimer is required for the early incorporation of the 5S RNP into the pre-ribosome [1,161,162,163,164]. Immediately after incorporation at the nucleolar maturation stage, the 5S RNP complex is presumably loosely associated with the particle and cannot be resolved by cryo-EM [31]. As soon as it is clearly visible on nucleoplasmic pre-60S particles [32,33], the 5S RNP adopts a very different orientation compared to the mature ribosome. To reach its final position, the 5S rRNA has to be rotated by ~180° [27,32,33,35,142]. This large-scale rearrangement has been proposed to be initiated by binding of Rix1 and has already taken place on particles containing both Rea1 and Rix1 [27]. A structure of the particle from a *rix1* mutant unable to recruit Rea1 has shown that the Rix1ΔC-Ipi1-Ipi3 complex is only loosely bound and contacts the tip of the 5S RNP in its unrotated form together with Rpf2–Rrs1 [27]. It has therefore been proposed that rotation of the 5S RNP may be triggered by destabilization of contacts between the 5S RNP and Rpf2–Rrs1 upon binding of the Rix1 complex [27]. Interestingly, the particle still was found to contain the foot structure, suggesting that ITS2 processing and rotation of 5S RNP might be temporally tightly linked. The binding site of the Rpf2–Rrs1 heterodimer, which stabilizes the immature 5S RNP position, partially overlaps with the fully inserted Rix1 sub-complex [27,165]. Consistently, Rpf2-Rrs1 is not found on particles containing the Rix1 sub-complex and Rea1. As recently described, the maturation factor Cgr1 stabilizes the mature conformation of the 5S RNP after rotation [165], which would allow Rix1 to stabilize and recruit Rea1. During this transition, Rsa4 also changes its conformation on the pre-ribosome and could assist Rea1 with adopting its active state on the pre-ribosome with the MIDAS domain attached to the AAA ring [27,128]. The coordination of 5S RNP remodeling, ITS2 processing, and binding of Rea1 is still not fully resolved and holds plenty of interesting questions for future research.

Downstream of 5S RNP remodeling, the activity of Rea1 has been thought to be linked to the next proposed ribosomal maturation checkpoint: the replacement of the GTPase Nug2/Nog2 by the export adaptor Nmd3 [129]. Due to significant overlap of the binding sites of these two proteins, Nog2 has to be released before the particles can be made ready for transport. Genetic interactions and biochemical data have suggested a tight functional coordination between ATP-dependent Rsa4 removal triggered by Rea1 and the subsequent GTP-dependent release of Nug2 [129]. However, this view has been challenged by recent structural data of late nuclear pre-60S particles that contain Nog2 and Nsa2 but lack Rsa4 and Rea1 [35]. Therefore, the release of Rsa4 by Rea1 must be an earlier event that is followed by dissociation of Nog2 and Nsa2. The cascade of linked steps during ribosome maturation is executed in an extremely narrow time window and thus it remains challenging to reliably derive the precise order of events and the tight network of direct functional linkages. Strategies to overcome these obstacles include the use of specific low molecular weight inhibitors that halt the maturation cascade at specific stages. However, specific inhibitors are only available for very few AAA-ATPases including *S. cerevisiae* Drg1 [143,166,167,168] and *S. pombe* Mdn1 [128,130] (recently reviewed in [169]).

#### 3.2.5. Inhibitor-Based Analysis of Mdn1

The accumulating data for *S. cerevisiae* Rea1 as well as human, plant, and *S. pombe* Mdn1 have provided more and more insights into the function of these AAA machines during pre-60S maturation, revealing parallels and differences between the species. Recently, a class of cell-permeable inhibitors targeting *S. pombe* Mdn1 (spMdn1) has been discovered from a chemical synthetic lethality screen [130] and further characterized by cryo-EM [128] (recently reviewed in [169]). These Rbins (short for Ribozinoindoles) have allowed snapshots of trapped intermediate states of spMdn1 during the ATPase cycle as described above [128] and have aided the study of the function of spMdn1 in vitro and in vivo [130].

Treating *S. pombe* with Rbins has been found to result in mislocalization of GFP-tagged Rpl2501 (uL23/L25) from the cytoplasm to the nucleolus but not reporters of the 40S subunit. This is indicative of a blocked export defect that is specific to the large ribosomal subunit [130]. Moreover, treatment with Rbin-1 has been seen to result in processing defects of the 27S and the 7S pre-rRNA [130], which is reminiscent of the defects observed for the genetically impaired Rea1 function in *S. cerevisiae* [83]. Interestingly, inhibition of spMdn1 has been observed to result in a transient accumulation of Rsa4 (but not Ytm1) on purified Rix1-particles after 15 min of treatment with Rbin-1 or using an Mdn1 ts-mutant [130]. The transient increase on the particle has been found to be followed by a reduction of both proteins after prolonged treatment. This two-phased pattern has been interpreted as a further indication of the involvement of Rea1 at more than one maturation step analogous to the situation in *S. cerevisiae*. Consistent with this hypothesis, spMdn1 interacts with both Ytm1 and Rsa4, but the interaction with Ytm1 appears weaker in *S. pombe* [130]. In addition, spMdn1 has been proposed to be connected not only to nucleoplasmic Rix1-associated pre-60S particles but also to nucleolar Nsa1-particles. Treatment with Rbins has been found to alter the composition of the early Nsa1-particle (e.g., leading to reduced amounts of Rix7), and therefore spMdn1 has been proposed to contribute to the assembly of these particles [130]. The lack of detection of Rix7 in the Nsa1-particle led to the hypothesis that incorrect assembly of the Nsa1-particle is the primary defect upon Mdn1 inhibition, with alterations at the later Rix1 particle as a secondary effect after prolonged time [130]. However, due to the recycling of maturation factors, disturbances of the ribosome biogenesis pathway are prone to also cause secondary effects upstream of the actual site of inhibition, as shown for inhibition of Drg1 [143,167,170]. Thus, in general the read-outs after inhibitor treatment have to be evaluated with care to correctly recognize and interpret rebound effects and differentiate between primary and secondary defects in the maturation cascade.

In addition to studying the cellular function of spMdn1, Rbins have been used to address the role of its ATPase activity for remodeling [130]. Treatment with 1µM Rbin-1 has been found to lead to a ~40–50% reduction of spMdn1 ATPase activity in vitro and a severe growth defect in vivo [130]. The exact binding site of the Rbins is still under investigation; however, these experiments have suggested that partial inhibition of the ATPase activity of Mdn1 is sufficient to disturb its functionality. It remains unknown how the individual AAA domains contribute to the release mechanism and whether Rbin treatment results in a general inhibition of all functional AAA domains of spMdn1 (AAA2-5) or only a subset of the AAA modules [128,130]. Interestingly, exchanges causing resistance to Rbins cluster in AAA3 and AAA4 and suggest that the nucleotide binding pocket formed between these two AAA domains is directly affected by inhibitor binding [128]. If the AAA3-AAA4 nucleotide binding pocket was the only site of inhibition this would suggest that it contributes ~50% of the overall ATPase activity of Rea1. Additional experiments utilizing an ATP-hydrolysis-deficient mutant within AAA5 (Walker B mutant) have shown drastically reduced ATPase activity (85% reduction compared to wildtype activity), and yet have still been inhibited by Rbins by ~40% [128]. Based on this strong effect, Chen and coworkers have speculated that blocked ATP hydrolysis in one domain might allosterically affect the activity of the other functional catalytic domains of Mdn1 (AAA2–AAA4) which is consistent with coordinated nucleotide binding and hydrolysis in other AAA-ATPases. Whether the inhibitor exclusively blocks ATP hydrolysis in one or a subset of the AAA domains or affects structural transitions of the remodeling reaction awaits further research.

### 3.3. Drg1: AAA-ATPase Ante Portas

#### 3.3.1. Structural Characteristics of Drg1

The gene coding for Drg1 was first identified during a screen for *S. cerevisiae* mutants resistant to the heterocyclic boron-containing inhibitor diazaborine [168]. Biochemical, genetic, and cell biological experiments have subsequently demonstrated that Drg1 is the direct and exclusive target of this compound in yeast [166,167,168,170,171,172]. *S. cerevisiae* Drg1 is evolutionarily related to p97/Cdc48, with ~50% sequence identity of the AAA-domains [85,87]. Although the NTD of Drg1 shows lower sequence conservation, it is proposed to be structurally similar to the p97 NTD with a bipartite sub-domain organization needed for substrate and cofactor interactions [26,84,87]. Flanking the three major domains (Figure 9A), Drg1 carries N-terminal and C-terminal extensions (amino acids 1–30 and 771–780, respectively) that are essential for growth. Deleting, for example, only the first 28 amino acids of Drg1 renders the protein non-functional (Bergler lab, unpublished results). These potentially rather flexible and unstructured extensions could be involved in substrate interactions (N-terminal extension) but might also be needed for the interaction with additional factors or post-translational modifications as shown for an unstructured C-terminal extension of p97/Cdc48 which serves as a phosphorylation site [173,174].

The function of the human orthologue of Drg1, SPATA5, is still unclear. Drg1 and SPATA5 share ~48% sequence identity, with the D2 domain being the most conserved region. Early reports linked SPATA5 to spermatogenesis [175] as well as Aurora kinase B extraction in *C. elegans* [176] and depleting SPATA5 has been found to affect mitochondria and neurons [177]. However, siRNA knock-down of SPATA5 has showed a pronounced 47S and 32S pre-rRNA accumulation, strongly suggesting a role in ribosome biogenesis [7]. Recessive mutations in SPATA5 have been recently linked to developmental delay, hearing loss, epilepsy, a microcephaly phenotype, and mental disorders [177,178,179,180,181]. Similar phenotypes are observed with mutants in human ribosome biogenesis genes and ribosomal protein genes [182]. The phenotypes observed in individuals carrying pathogenic SPATA5 alleles might therefore arise from decreased proliferation of neuronal progenitor cells due to a block in ribosome biogenesis as previously proposed for other ribosome biogenesis defects [182]. This evidence suggests that SPATA5 fulfils a conserved role in ribosome biogenesis.

#### 3.3.2. Drg1 Initiates the Cytoplasmic Pre-60S Maturation Cascade

The final assembly stage of the large ribosomal subunit is accomplished by multiple steps in the cytoplasm. After passing through the nuclear pore complexes, the pre-ribosomes associate with Drg1 (Figure 9B), triggering the release of the shuttling factor Rlp24 [85,170,183]. Rlp24 accompanies pre-ribosomes from the nucleolus and thereby acts as a placeholder for the r-protein L24 (eL24), which can only be incorporated after removal of Rlp24 [85,184]. After release, Rlp24 is recycled back into the nucleolus so that it can associate with freshly synthesized pre-ribosomes and serve its role in early steps of pre-ribosome maturation [170,185]. Cryo-EM structures of purified pre-ribosomal particles have shown that Rlp24 is closely intertwined with the maturation factors Nog1 and Bud20 [31,32,33,34]. Bud20 supports nuclear export of the pre-60S subunit and has to be efficiently shuttled back into the nucleus to facilitate the export of succeeding pre-ribosomes [186,187]. The GTPase Nog1 contains an N-terminal four-helical-bundle domain positioned in the A-site [33]. An additional long-expanded helical domain of Nog1 spans over half of the large subunit up to the solvent side. The final Nog1 helix inserts deep into the PET [28,31,33]. After release of Nog1, the PET is occupied by Rei1. The loading of Rei1 is linked with the release of Arx1 [188]. Finally, in late cytoplasmic pre-60S particles, Rei1 is substituted by Reh1, which resides in the PET till the very end of the maturation process [30]. Thus, the PET is sequentially occupied or probed by various assembly factors during the maturation cascade to ensure correct formation and protection of the tunnel.

The tight entanglement of Nog1 and Rlp24 has raised the question of whether the release of both factors is directly linked to the activity of Drg1. In vitro, however, Nog1 stays on the Arx1-particle after Rlp24 is already released [85], contradicting a coupled release of both proteins in one step. Still, the removal of Nog1 seems to be temporally tightly linked to the release of Rlp24 and can only occur when Drg1 is functional [85,166,183]. Due to the tight entanglement of these proteins, extraction of the C-terminal tail of Nog1 from the PET has been proposed to be triggered by the removal of Rlp24, while the N-terminus of Nog1 is only removed after GTP hydrolysis [189]. This hypothesis awaits testing by structural analysis. Nevertheless, dissociation of Rlp24 is a strict prerequisite for all downstream maturation steps in the cytoplasm [85,170,183]. These final adjustments include release of export factors, loading of remaining ribosomal proteins and finalization of the PTC, the catalytic core of the ribosome [30,183,185,190,191,192,193,194,195]. Thus, the active removal of Rlp24 by Drg1 triggers a major remodeling cascade of the pre-60S subunit shortly after export, paving the way for all downstream reactions (Figure 9B).

Intriguingly, a fragment of the FG-repeat-containing nucleoporin Nup116 interacts with the NTD of Drg1 and is required for the release of Rlp24 in vitro [85]. This connection to a nucleoporin corroborates the hypothesis that the release of Rlp24 by Drg1 is tightly linked to the export of the pre-60S particle through nuclear pore complexes (NPC). Nup116 is not needed to recognize the substrate Rlp24 and therefore it might act as an additional anchoring point at the NPC to generate tension for the mechanical extraction of Rlp24. Although structural information about the release event is lacking, overall conservation of the protein domains of Drg1 and its pore loop residues suggest that processing could also involve at least partial threading of the substrate protein into the central pore. However, since Rlp24 has to be recycled to allow ongoing formation of ribosome biogenesis, a full unfolding of the shuttling protein seems counterproductive. Due to the missing structural data, the detailed mode of substrate processing and whether the substrate is thereby unfolded awaits further investigation. Nevertheless, thorough genetic and biochemical investigations have already allowed informative insights.

#### 3.3.3. Substrate Recognition and Processing by Drg1

The compact globular N-terminal part of Rlp24 acts as an rRNA-interacting domain and is tightly buried in the pre-ribosome [28,31,32,33]. This N-terminal domain is followed by a long α-helical stretch and an exposed flexible C-terminal domain (C-domain). The major part of Rlp24 is highly similar to L24 (eL24); however, the C-domain is an exclusive feature of the maturation factor [32,33,85,183]. The C-domain contains the last ~50 residues of Rlp24, which are rich in charged amino acids. Although the C-domain is presumably unstructured, it can specifically recruit Drg1 via its NTD with a binding affinity in the nM range [85]. The exclusive presence of this C-domain explains why Drg1 is only found on Rlp24-containing pre-ribosomes but not on particles where L24 (eL24) is already loaded [35,85,184]. Intriguingly, the interaction with the Rlp24 C-domain not only recruits Drg1 to the particle but also stimulates ATP hydrolysis in both of its AAA domains [85]. ATP hydrolysis in this activated state seems to provide the energy to actively drag Rlp24 out of the pre-ribosome. This mechanism ensures that ATP is only hydrolyzed in the presence of a bound substrate. Similar cases of substrate- or cofactor-dependent stimulation of ATPase activity have been observed for other AAA-ATPases [196,197,198,199,200,201,202,203,204]. Substrate- or cofactor-dependent stimulation seems to represent a conserved regulatory feature of related AAA-ATPases. However, with few exceptions (e.g., Torsin [196]), it is unclear how binding of a substrate or co-factor enables the AAA domain(s) to hydrolyze ATP more effectively. As described above, in the case of Rea1, ATP hydrolysis could be stimulated at the ribosome by displacing the auto-inhibitory H2α insertion from the ring [127,128].

As Type II AAA-ATPases, Rix7 and Drg1 carry a total of 12 functional ATPases sites per hexamer and it is still one of the most prevalent questions as to how these catalytic sites cooperate and contribute to substrate processing. The assignment of the two domains has been not only studied by using mutant variants deficient in ATP binding (Walker A mutants) or ATP hydrolysis (Walker B mutants) but also by the use of the inhibitor diazaborine which specifically blocks ATP hydrolysis in the D2 domain of Drg1 [68,85,87,101,166]. The interaction with Rlp24 requires loading of ATP into the D1 domain of Drg1 as a prerequisite for hexamerization of the AAA-ATPase. Accordingly, mutant variants of Drg1 unable to form hexamers (e.g., the temperature sensitive *drg1-18* variant) fail to bind to Rlp24 and therefore cannot associate with the pre-60S particle [85]. Similarly, a mutant version of Drg1 unable to load ATP into D1 (D1-Walker A mutant) which does not form hexamers fails to interact with Rlp24 [85,87,172]. Interestingly, a Drg1 variant that can load ATP in D1 but is unable to hydrolyze this bound nucleotide (D1-Walker B mutant) shows much stronger binding to the Rlp24 C-domain compared to the wildtype protein, suggesting that ATP hydrolysis in D1 dissociates the AAA-ATPase from its substrate [85,166]. Both nucleotide binding and hydrolysis in D1 modulate substrate interaction of Drg1. By contrast, a mutant variant unable to bind nucleotides into D2 (D2-Walker A mutant) is still able to form hexamers and shows similar association kinetics to the Rlp24 C-domain as wildtype Drg1 [87]. Due to the missing nucleotide in D2, this mutant variant dissociates much faster from the Rlp24 C-domain. This suggests that after initial recognition of the Rlp24 C-domain by hexameric Drg1, ATP has to also be loaded into the D2 domain to allow stable interaction with the substrate. Together, ATP binding in both AAA domains of Drg1 regulates and coordinates the transient interaction with its substrate Rlp24.

The release of Rlp24 from the Arx1-particle has been recapitulated in vitro and strictly requires ATP hydrolysis in the D2 domain of Drg1 [85,166]. This is obvious by the failure of the D2 ATP-hydrolysis-deficient Drg1 (D2-Walker B) mutant to release Rlp24 in vitro and a pronounced dominant negative effect upon overexpression of this mutant protein [85]. By contrast, the Drg1 D1-Walker B mutant still shows ~60% in vitro release compared to the wildtype protein ([85,166]; Bergler lab, unpublished results). Accordingly, although not strictly required for survival, ATP hydrolysis in D1 also contributes to efficient in vitro release. Thus, ATP hydrolysis in both domains is needed for efficient release and recycling of Rlp24 from the pre-ribosome. In summary, both AAA domains of Drg1 collaborate for optimal substrate processing but contribute differently. In general, a division of labor between the AAA domains is a shared feature of type II AAA-ATPases, e.g., NSF [79], Cdc48 [72,73], and Rix7 [68], but it is not strictly determined what role has to be played by the D1 and the D2 domains.

#### 3.3.4. Capturing Ribosome Biogenesis Dynamics with the Drg1-Inhibitor Diazaborine

The observation that release of Rlp24 initiates the cytoplasmic maturation cascade was crucial to understanding how inhibiting Drg1 by diazaborine results in entrapment of all known shuttling proteins on pre-60S particles shortly after nuclear export ([85,143,166,170,183,187]; Bergler lab, unpublished results). Diazaborine specifically inhibits ATP hydrolysis in the D2 domain, while ATP hydrolysis in the D1 domain is not affected [166]. Indeed, many mutations leading to diazaborine resistance result in exchanges clustering around the D2 nucleotide-binding pocket and cause reduced binding of the drug. This suggests that the compound directly binds into the D2 domain and thereby blocks ATP hydrolysis in this site [166]. Interestingly, weak acid stress also seems to affect the activity of Drg1 as recent PAR-TRAPP data from the Tollervey lab has shown increased levels of Rlp24 containing pre-ribosomes and decreased levels of all downstream intermediates upon exposure to sorbic acid [205]. Intriguingly, this marked response mirrors the effect of diazaborine and likely arises from a pronounced pH dependency of the D2 domain of Drg1, possibly as a consequence of the cysteine residue present in the Walker A motif in D2 (Bergler lab, unpublished results).

Similarly to chemical inhibition of Drg1, overexpression of a Rlp24 construct lacking the C-domain, which fails to recruit Drg1, also results in accumulation of a pre-60S population shortly after export. This recently visualized particle population contains all shuttling proteins as well as the export adaptor Nmd3 and exhibits a closed L1 stalk conformation [34]. In the same study, a late nuclear population of particles has also been detected that contains Nsa2 and Nog2 but lacks Rsa4 and the export adaptor Nmd3. Interestingly, very similar particle populations have been independently isolated after treatment of cells with diazaborine (Warren, Bergler labs, unpublished results). The smaller late nuclear population could arise from a failure to recycle the export adaptor Nmd3, which is trapped at early cytoplasmic particles due to the failure to release Rlp24 (Bergler lab, unpublished results). Thus, efficient recycling of shuttling factors is crucial for the efficient production of new ribosomes since there is presumably no significant pool of free factors under normal conditions. Therefore, by blocking recycling of shuttling proteins from the cytoplasm, diazaborine treatment also rapidly results in earlier 27S pre-rRNA processing defects in the nucleolus due to shuttling factor depletion at this early stage. Additionally, as a further consequence of this blockage, later joining ribosomal proteins (e.g., L24 (eL24) and L10 (uL16)) and maturation factors accumulate in free form and eventually precipitate which results in activation of the heat shock transcription factor HSF1 [206,207].

Discovering that inhibition of late steps also rebounds on very early events has also revealed the dynamics and complexity of the ribosomal maturation cascade and the unique advantage of targeted chemical inhibition. The ability to use chemical inhibition of AAA-ATPases and other key players as a tool that acts almost immediately allows unprecedented resolution of the occurring effects. The effects on protein composition and the pre-rRNA processing state of different pre-60S particles upon diazaborine treatment can be observed after very short times (<2 min). Such timely resolution cannot be achieved by genetic or classical biochemical approaches for the depletion of proteins but is indispensable to disentangle primary and secondary effects of an inhibited ribosome biogenesis pathway. This highlights the power of low molecular weight inhibitors not only in inhibition of particular enzymes like the AAA-ATPases Drg1 or Rea1 but also in dissecting the extremely interwoven ribosome biogenesis pathway. Accordingly, analyzing ribosome biogenesis with chemical probes will provide fresh insights into the dynamics of the pathway and lead to a novel understanding of the process [11,143].

## 4. Concluding Remarks and Perspectives

With recent advancements in cryo-EM and the discovery of new chemical inhibitors there has been an explosion of new information about the structure and function of the three AAA-ATPases required for maturation of the large ribosomal subunit. We now have near-atomic resolution views of the motor domains of Rix7 and Rea1/Mdn1 and are beginning to understand how ATP hydrolysis is coupled with remodeling of the large ribosomal subunit. Cryo-EM structures of these ATPases in action have revealed similarities with other AAA+ family members and have also identified unique features. For example, Rix7 and Drg1 draw many parallels with the well-studied Cdc48/p97, such as having a shared mechanism of unfolding; however, Rix7 has a distinct NTD and unique post-α7 insertion important for ribosome assembly. Likewise, the Rea1 concatenated AAA ring is structurally similar to dynein, yet Rea1 does not appear to utilize a dynein-like power-stroke mechanism to drive assembly factor release. Moreover, work with the inhibitors Rbin-1 and diazaborine has led to novel insights into the dynamics of ribosome assembly.

Despite recent advances in the structure and function of Rix7, Rea1, and Drg1, our understanding of the roles these AAA-ATPases play in ribosome maturation is far from complete. One of the biggest outstanding questions is how do these AAA+ machines engage with pre-60S particles at the appropriate stage in assembly? Aside from a structure of *S. cerevisiae* Rea1 bound to late stage nuclear pre-60S particles we do not have any structures of Rix7 and Drg1 bound to pre-60S particles or a structure of Rea1 on the pre-60S before Ytm1 release. These structures are greatly anticipated and will no doubt provide a wealth of insight into how these ATPases are recruited to pre-60S particles and what triggers ATP hydrolysis. Another major outstanding question about Rix7, Rea1, and Drg1 centers on the function of these motors across eukaryotes. Do the human homologues NVL2, MDN1, and SPATA5 play analogous roles in ribosome assembly as their yeast counterparts? Studies with Rix7 and its mammalian homologue NVL2 suggest that Rix7 and NVL2 perform similar functions in 60S maturation. However, recent work has revealed that vertebrate homologues of NVL2 associate with the nuclear exosome complex through an Mtr4-binding motif absent in lower eukaryotes. This suggests that vertebrate NVL2 may have additional functions in pre-rRNA processing. Thus, it will be interesting to see if MDN1 and SPATA5 have also acquired additional functions in higher eukaryotes. Many ATPases, such as Cdc48/p97, have diverse cellular roles but it is not known if Rix7, Rea1, and Drg1 have any additional remodeling targets beyond pre-ribosome particles. Studies to address substrate specificity, single molecule remodeling/unfolding experiments and the development of loss-of-function mutants are all needed to tackle this important question. Finally, ribosome assembly has emerged as a new target for anti-cancer therapeutics, and thus, Rix7, Rea1, and Drg1 could be potential therapeutic targets. Therefore, the development of new AAA inhibitors targeting these fundamental enzymes might not only further enhance our understanding of biological function but may also prove useful in cancer treatment.

## Figures and Tables

**Figure 1 biomolecules-09-00715-f001:**
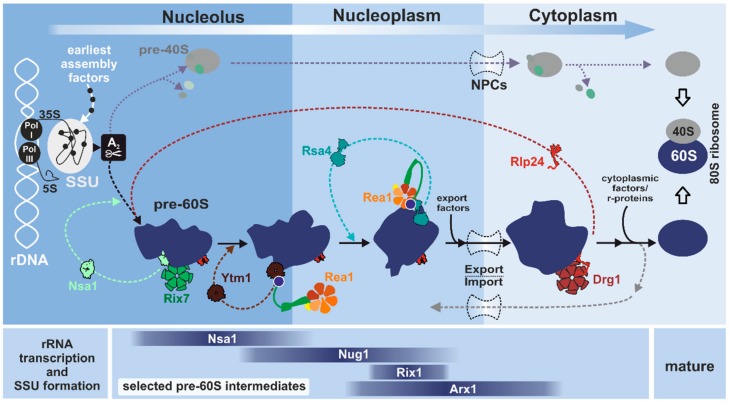
Eukaryotic ribosome biogenesis—a sophisticated assembly line. Ribosome biogenesis begins with the transcription of the ribosomal RNA (rRNA) in the nucleolus. At this stage, the first maturation factors assemble and form the earliest pre-ribosomal particle (small subunit (SSU)-processome). After endonucleolytic cleavage of the primary rRNA transcript, pre-40S and pre-60S particles go through separated maturation pathways. Numerous (re)-assembly, rearrangement, and modification steps occur in pre-ribosomal particles as they transit from the nucleolus to nucleoplasm and finally to the cytoplasm where they obtain their mature form. Pre-ribosomes associate with different maturation factors that define distinct maturation intermediates, indicated in the lowered box (Nsa1, Nug1, Rix1, and Arx1). The yeast AAA-ATPases Rix7 (green), Rea1 (multi-colored) and Drg1 (red) associate at different stages of assembly with pre-60S subunits and catalyze the release of specific maturation factors, which are recycled and join freshly produced pre-ribosomes. Only selected particles as well as maturation factors are displayed and the rRNA is omitted except for the SSU. The individual maturation events triggered by these AAA-ATPases will be described in detail in later sections. NPCs: Nuclear pore complexes.

**Figure 2 biomolecules-09-00715-f002:**
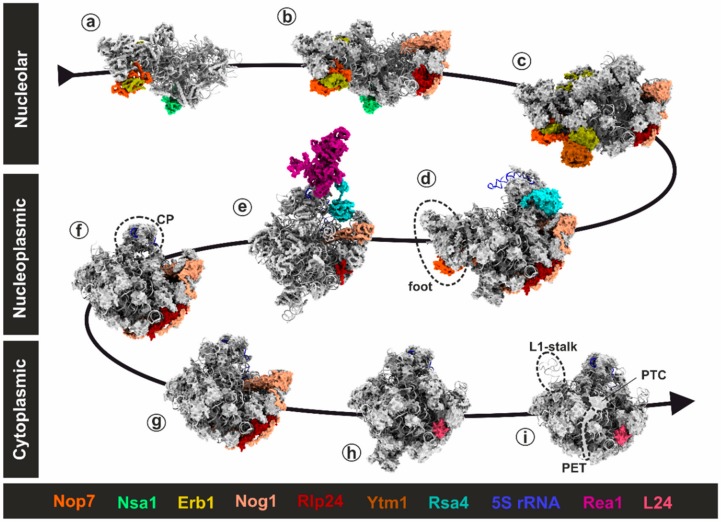
Transformation of pre-60S particles. Selected cryo-EM structures of yeast pre-60S particles highlight the structural and conformational transformation during the maturation cascade from the nucleolus to the cytoplasm. The rRNA undergoes manifold rearrangements, maturation factors (colored legend) temporarily associate with the pre-ribosomal particles, and the r-proteins are stepwise-incorporated. The characteristic foot structure is removed in the nucleoplasm and hallmark structures of the large subunit take on their final shape, including the central protuberance (CP), the polypeptide exit tunnel (PET), the peptidyl transferase center (PTC) and the L1 stalk. a–c: PDB ID 6EM3, 6EM1, and 6ELZ [31]. d: PDB ID 3JCT [33]. e: PDB ID 5JCS [27]. f–g: PDB ID 6N8J and 6N8K [35]. h–i: PDB ID 6RZZ and 6QT0 [30].

**Figure 3 biomolecules-09-00715-f003:**
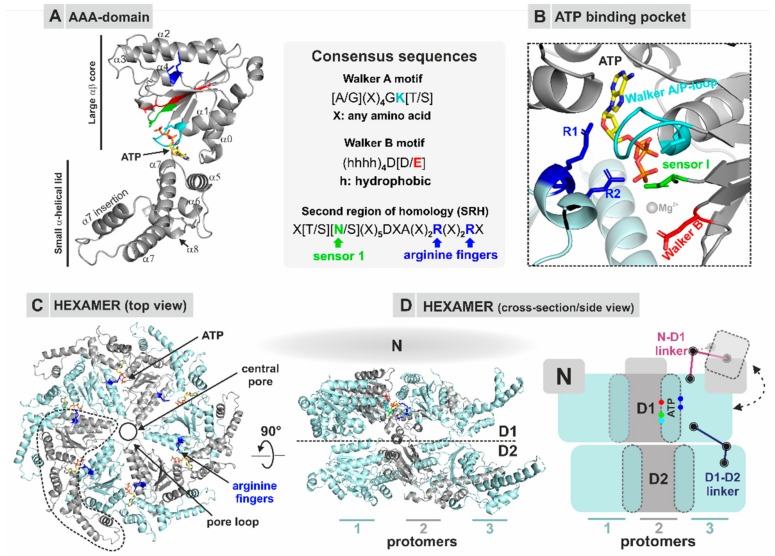
General architecture of type II AAA-ATPases. (**A**) D1 AAA module of the *Chaetomium thermophilum* type II AAA-ATPase Rix7 (PDB ID 6MAT), composed of a large αβ core and a small α-helical lid domain. Important structural elements (Walker A/B, R-finger, and sensor I) are colored as indicated. Each type II AAA-ATPase protomer contains two AAA modules (D1 and D2). (**B**) Magnification of the nucleotide binding pocket formed at the interface between two protomers. Nucleotide-sensing arginine fingers in the second region of homology (SRH) are provided by the neighboring protomers and allow communication of the nucleotide binding state. (**C**) D1 ring of Rix7 in top view. The monomers are colored alternately in gray and cyan. The ring-shaped hexameric assembly allows the formation of a central channel lined by conserved pore loops involved in substrate threading. The D2 ring is not depicted. (**D**) Cross-section of the Rix7 hexamer in side view (cartoon and schematic), showing the D1 and D2 ring. The flexible N-terminal domain (NTD) depicted in the scheme is not resolved in the shown cryo-EM structure of Rix7. The individual domains are connected by flexible linker sequences (only shown for protomer 3) that allow movements of the domains dependent on the ATPase cycle (e.g., up/down movement of the N-terminal domain) as well as communication between the domains.

**Figure 4 biomolecules-09-00715-f004:**
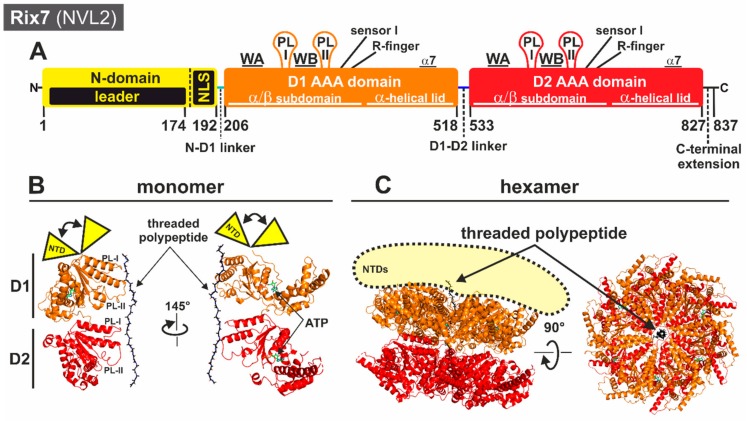
Domain organization and structure of the AAA-ATPase Rix7. (**A**) Rix7 contains two AAA modules D1 and D2 and an N-terminal domain per monomer. Annotation of the amino acid positions refers to the yeast proteins. The Rix7 NTD contains a nuclear localization sequence (NLS) that recruits the protein to the nucleus. A more detailed depiction of the Rix7 domain organization can be found in [68]. (**B**,**C**) Pore loops in each AAA module contain residues that form the inner lining of the central channel and are involved in the translocation of a threaded substrate [67]. The NTD is proposed to be dynamic and flexible and was not clearly localized in the structure of the substrate-bound *C. thermophilum* Rix7 complex (PDB ID: 6MAT). WA: Walker A motif. WB: Walker B motif. PL-I: Pore loop 1. PL-II: Pore loop 2.

**Figure 5 biomolecules-09-00715-f005:**
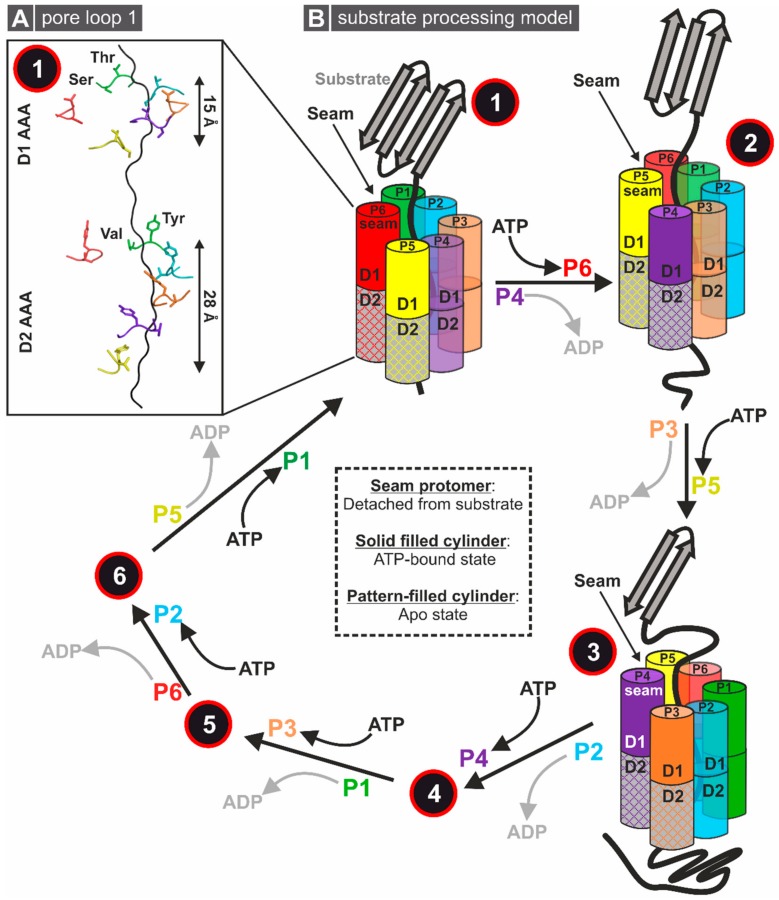
Working model for substrate translocation by Rix7. (**A**) Positions of pore loops 1 (PL-I) surrounding the bound substrate (threaded polypeptide, black cartoon). Individual protomers are colored as follows. P1: green, P2: cyan, P3: orange, P4: purple, P5: yellow, and P6: red. PL-I residues Thr276 and Ser277 in D1 domain and Tyr575 and Val576 in D2 domain, are shown as sticks. (**B**) Processive translocation model: Step 1: P1 binds the substrate at the top of the spiral configuration, followed by the next four protomers (P2–P5). The final protomer, P6 or seam, does not contact the substrate and is set back from the central channel. Although P5 contacts the substrate, the nucleotide binding site of the P5 D2 domain is empty, suggesting it has already hydrolyzed ATP and released ADP. Step 2: As ATP binds to the P6 D2 domain, P6 is enabled to reengage with the substrate and move to the location at the top of the spiral configuration. P5 then disengages from the substrate and takes over the position as the new seam protomer. ATP hydrolysis in the P4 D2 domain provides the energy to translocate the substrate. Steps 3–6: The process repeats sequentially on the D2 domains within the Rix7 hexamer through successive rounds of ATP hydrolysis. This working model is based on cryo-EM studies of the substrate bound *C. thermophilum* Rix7 (PDB ID: 6MAT). ATP-bound protomers are illustrated as solid-filled cylinders, whereas protomers without nucleotides are illustrated as pattern-filled cylinders.

**Figure 6 biomolecules-09-00715-f006:**
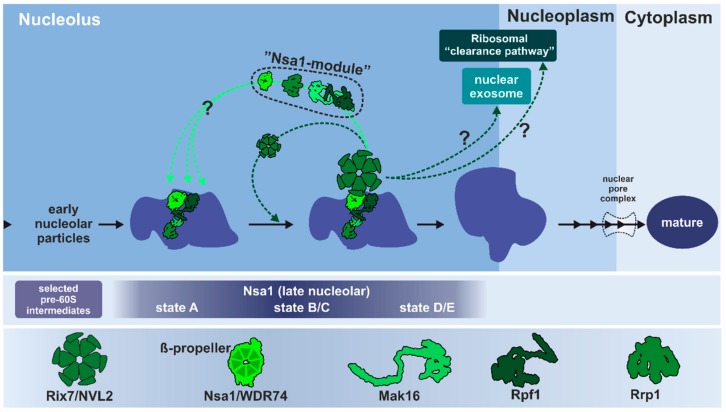
Remodeling of the nucleolar Nsa1-particle by Rix7. *S. cerevisiae* Rix7 joins late nucleolar Nsa1-particles to trigger the release of the β-propeller protein Nsa1, which is part of assembly factor sub-complex (“Nsa1-module”) [31]. Timely release of these factors is considered a prerequisite for correct folding of the pre-rRNA as well as the formation of the PET, which at this early stage is occupied by Rpf1. It is, however, not determined if these four proteins join the particle together and also if they are released together at the exact same stage. As discussed in this section, Rix7 could potentially also be connected to a ribosomal clearance pathway as part of the ribosomal quality control. Activity of the mammalian Rix7-orthologue nuclear VCP-like protein 2 (NVL2) is needed for the assembly of the nuclear exosome which also contributes to 60S maturation.

**Figure 7 biomolecules-09-00715-f007:**
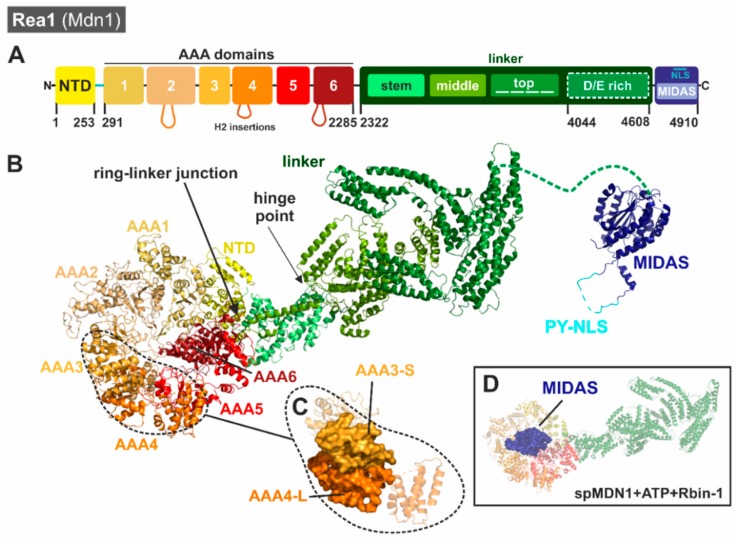
Structure of Rea1/Midasin 1 (Mdn1). (**A**) Rea1/Mdn1 is composed of a short N-terminal domain, followed by six adjacent AAA modules and the tail-like extension, all combined in one polypeptide chain. The linker is composed of a structured stem followed by a flexible D/E-rich region and connects the substrate-binding metal-ion-dependent adhesion site (MIDAS) domain to the AAA ring. The MIDAS domain also contains a PY-NLS needed to import Rea1 to the nucleus [126]. The six AAA modules that form the hexameric ring vary in size and partially contain additional helix insertions that can be involved in the interaction with the pre-ribosome [27]. (**B**) The AAA-domains and linker are taken from the cryo-EM reconstruction of *S. cerevisiae* Rea1 PDB ID: 6I26 [127] and the MIDAS domain from the recent crystal structure of the *C. thermophilum* MIDAS PDB ID: 6QT8 [126]. (**C**) Magnified interface created by two adjacent Rea1 AAA domains (AAA3-S and AAA4-L). (**D**) Weak density for MIDAS in contact with the ring detected in the cryo-EM reconstruction of *S. pombe* Mdn1 in the presence of ATP + Rbin-1 (PDB ID 6ORB) [128].

**Figure 8 biomolecules-09-00715-f008:**
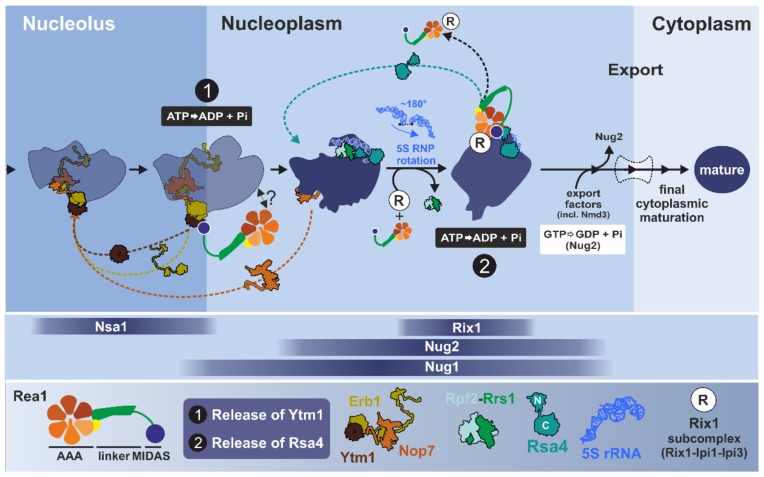
Two-step remodeling of pre-60S particles by Rea1. Yeast Rea1 is proposed to associate with the pre-60S particle at two distinct stages, these being first at the transition from the nucleolus to the nucleoplasm and secondly at a later stage of nucleoplasmic maturation. Both maturation stages are associated with the ATP-dependent release of maturation factors (Ytm1-Erb1 and Nop7 as well as Rsa4) and structural rearrangements of the pre-ribosome (e.g., rotation of the 5S RNP). In the nucleoplasm, the Rix1 sub-complex binds to the particle and recruits Rea1 via direct interaction between Rix1 and Rea1. Downstream, the release of Rsa4 is followed by GTP-dependent dissociation of Nug2 and loading of Nmd3, which is a prerequisite to render the particle export competent.

**Figure 9 biomolecules-09-00715-f009:**
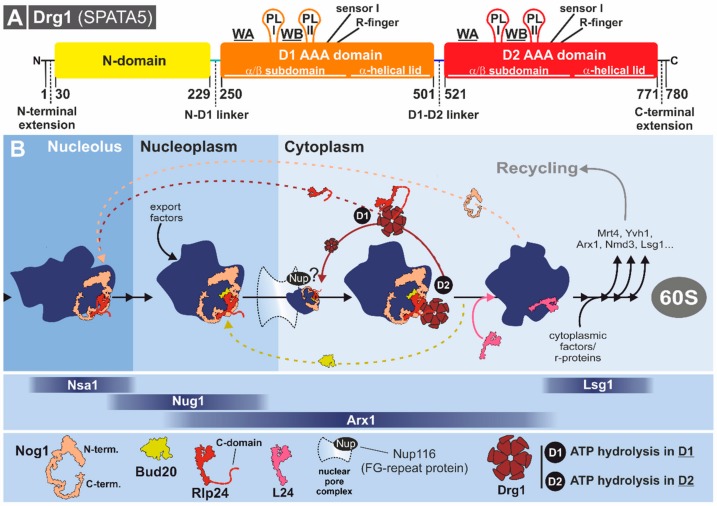
Structure and Function of Drg1. (**A**) Domain organization. The very N-terminal extension of Drg1 (spermatogenesis-associated 5 (SPATA5) in humans), which is absent in the related AAA-ATPase p97, is essential for stimulation of the ATPase activity by Rlp24. Both AAA domains of Drg1 are catalytically active and contain putative arginine finger motifs. The pore loop motifs are partially conserved but no data are available yet as to whether substrate processing of Drg1 involves full or partial threading of the substrate Rlp24. (**B**) Drg1 initiates the cytoplasmic pre-60S maturation cascade in yeast. Drg1 is recruited to pre-60S particles immediately after their export into the cytoplasm. The flexible C-terminal domain of the shuttling maturation factor Rlp24 recruits hexameric Drg1 via its NTD and stimulates ATP hydrolysis in both AAA domains of Drg1 to initiate the release process. ATP hydrolysis in the D2 domain releases Rlp24 from the pre-60S particle, while ATP hydrolysis in D1 subsequently dissociates the substrate Rlp24 from Drg1. Release of Rlp24 initiates cytoplasmic pre-60S maturation and is therefore required for all downstream events, including release and recycling of export and shuttling factors including Nog1, Bud20, and Mtr4. Rlp24 is recycled back into the nucleolus and its place at the ribosome is taken by the ribosomal protein L24 (eL24). Subsequently, the final r-proteins can be incorporated. FG-repeat: Phenylalanine-glycine repeats.

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
