# Peer review of "Shaping the Nascent Ribosome: AAA-ATPases in Eukaryotic Ribosome Biogenesis"

_biomolecules, 2019, doi:10.3390/biom9110715_

Round 1

Reviewer 1 Report

biomolecules-625039

Shaping the nascent ribosome: AAA-ATPases in eukaryotic ribosome biogenesis

By Michael Prattes et al.

In their review article, the authors nicely introduce the reader to the field of ribosome biogenesis and to the world of AAA-ATPases in general. Structure and function of the three AAA-ATPases involved in eukaryotic ribosome biogenesis -Rix7 (NVL2), Rea1 (Mdn1) and Drg1 (SPATA5)- are well presented both in written words and descent figures.
The review is well written, the content is profoundly presented, quoting the relevant literature including the most recent.

Both experts and newcomers in the field of eukaryotic ribosome biogenesis will benefit from the reading of this well-made review.

Minor points:

Line 53: The abbreviation snoRNP (small nucleolar ribonucleoprotein) should be spelled out in parenthesis when used for the first time in the text.

Line 175: The authors mention novel “high-resolution cryo-EM structures”. The should include the references at the end of the sentence, if there are not too many.

Line 252: To be consistent with American spelling the word “hydrolyzable” should be used.

Author Response

Response to Reviewer 1:

We thank Reviewer 1 for the very positive comments on the manuscript. We have corrected all of the minor points raised by the reviewer in the revised manuscript.

Reviewer 2 Report

This manuscript from the Bergler and Stanley labs provides an updated review on three well-conserved AAA-ATPases involved in eukaryotic ribosome biogenesis, Rix7, Rea1 and Drg1 (yeast names). The manuscript is very pertinent for researchers specialised in the ribosome biogenesis/function field and attractive for a general audience. The manuscript is very nice, well-written and the different sections well documented. Importantly, it is easy to read. Given the fact that the ribosome biogenesis pathway is best known in the yeast Saccharomyces cerevisiae, most reviewed data concern particular aspects of this microorganism. However, when possible, the processes are explained in the broadest context of other eukaryotes, such as the fission yeast or mammals. The figures are very informative too.

I only have very few comments:

Although a real pain, I believe the authors should also use the novel nomenclature suggested by Ban et al. for naming ribosomal proteins. This could be for non-specialised readers, e.g. Rpl2501 is not obvious to be uL23/L25 from S. pombe (pag. 18, line 695). As most statements correspond to processes or steps only described in yeast, I recommend the authors to carefully revised the manuscript to always mention this fact all over the review. The Rvb1/2 proteins are not reviewed; together wih Rix7, Rea1 and Drg1 are the sole AAA-ATPases involved in ribosome biogenesis, thus, the second sentence of the abstract "Multiples AAA-ATPases…." can lead to the confusion that more than 4 proteins work in this process. Define MDA (I guess MegaDalton) in the abstract section. Page 1, line 41. As the Tafforeau et al. reference from Lafontaine lab is indeed included in the reference list, please cite it already herein together with the ref. Bohnsack and Bohnsack. Figure 3 is difficult to understand as such. In my humble opinion, a side view of the ring of Rix7 should be also shown in 1C to understand better the scheme show in 1D. Moreover, a citation of the pdb file for the structure shown is missing in the legend. Figure 4 legend. Please state that the protein is from Chaetomium. Authors should consider the possibility to make a summary table including few key aspects of the reviewed proteins: molecular weight, enzymatic activity, structure, yeast viable mutants, step where they work in ribosome biogenesis, factor targets, interaction with their targets, interaction with pre-60S particles, presence or not in cryo-EM particles, etc… In my opinion, this summary table would be nice for both specialised and aficionado readers.

Author Response

Response to Reviewer 2:

We thank Reviewer 2 for the support of the manuscript and the constructive comments. We followed the suggestions of the reviewer and made most of the requested changes to the manuscript. We updated Figure 3 to include a side view cross section of Rix7 and we added the PDB identifier and origin statements. We also updated the text to include the new nomenclature of ribosomal proteins. To avoid confusing readers the new nomenclature is shown in parenthesis. In addition, we carefully checked the manuscript for clarity for which organism the respective protein originates and included a respective statement when this was unclear. Finally we modified the abstract to make it clear that this manuscript is focused on the three AAA-ATPases directly involved in the remodeling of the pre-ribosomal particle itself (and not Rvb1/2).

While we appreciate the reviewer’s suggestion to add a summary table (MW, activity, yeast mutants, interaction with pre-60S particles etc) we feel all of this information is clearly presented to the readers in both the figures and text. For example, Figure 1 is an overview of where each AAA-ATPase functions in the pre-60S pathway.

Reviewer 3 Report

The review "Shaping the nascent ribosome: AAA-ATPases in eukaryotic ribosome biogenesis" summarizes the recent progress in our understanding of the essential AAA+ ribosome maturation factors Rix7, Rea1 and Drg1. The review is nicely structured, well written and should be published in "biomolecules".

I just have one very minor comment/suggestions:

-On page 15, line 552 the authors write: "Collectively, these structures suggest that Rea1 does not function by long-range motions within the linker like dynein but rather works by conformational changes within the AAA ring that lead to the binding and displacement of the MIDAS domain."

It is true that in the case of the AAA ring bound MIDAS domain nucleotide induced conformational changes of the ring might might be directly communicated to the MIDAS domain to remove Rsa4. For Rsa4 removal linker remodelling might indeed not be necessary. However, Ulbrich et al. 2009 have demonstrated that different linker conformations with respect to the AAA ring do exist and they still might be important for the removal of Ytm1. Perhaps the authors could add a sentence to that extend.

Author Response

Response to Reviewer 3:

We thank Reviewer 3 for the positive feedback and for drawing our attention to this open question about the linker domain of Rea1. We have included a respective statement regarding the linker conformation into the revised manuscript as requested.